# Functional Connectivity between Task-Positive Networks and the Left Precuneus as a Biomarker of Response to Lamotrigine in Bipolar Depression: A Pilot Study

**DOI:** 10.3390/ph14060534

**Published:** 2021-06-03

**Authors:** Marieke Martens, Nicola Filippini, Charles Masaki, Beata R. Godlewska

**Affiliations:** 1Department of Psychiatry, University of Oxford, Oxford OX3 7JX, UK; marieke.martens@psych.ox.ac.uk (M.M.); charles.masaki@mgh.harvard.edu (C.M.); 2Wellcome Centre for Integrative Neuroimaging, University of Oxford, Oxford OX3 9DU, UK; nicola.filippini@psych.ox.ac.uk; 3Oxford Centre for Human Brain Activity, Department of Psychiatry, University of Oxford, Oxford OX3 7JX, UK; 4Oxford Health NHS Foundation Trust, Oxford OX3 7JX, UK

**Keywords:** bipolar depression, treatment response, lamotrigine, resting-state fMRI, resting state functional connectivity, fronto-parietal network, dorsal attention network, default mode network, independent component analysis

## Abstract

Treatment of bipolar depression poses a significant clinical challenge. Lamotrigine is one of a few efficacious drugs, however, it needs to be titrated very slowly and response can only be assessed after 10–12 weeks. With only a proportion of patients responding, an exploration of factors underlying treatment responsivity is of paramount clinical importance, as it may lead to an allocation of the drug to those most likely to respond to it. This study aimed at identifying differences in patterns of pre-treatment resting state functional connectivity (rsFC) that may underlie response to lamotrigine in bipolar depression. After a baseline MRI scan, twenty-one patients with bipolar depression were treated with lamotrigine in an open-label design; response, defined as ≥50% decrease in Hamilton Depression Rating Scale (HAMD) score, was assessed after 10–12 weeks of treatment. Twenty healthy controls had a baseline clinical assessment and scan but did not receive any treatment. Fifteen out of 21 (71%) patients responded to lamotrigine. Treatment responsivity was associated with enhanced pre-treatment rsFC of the right fronto-parietal network (FPN) and dorsal attention network (DAN) with left precuneus. The lack of treatment response was additionally characterised by reduced rsFC: of the DAN with right middle temporal gyrus; of the default mode network (DMN) with left precuneus; of the extended sensory-motor area with areas including the left hippocampus/left amygdala and left subcallosal cortex/nucleus accumbens; and of the left FPN with left inferior temporal gyrus/occipital fusiform gyrus/lateral occipital cortex. The results suggest that preserved rsFC between the FPN and DAN, the networks involved in cognitive control, and the hub of the posterior DMN, the left precuneus, may be critical for good response to lamotrigine as an add-on treatment in patients with bipolar depression. The study also suggests a more general decrease in rsFC to be related to poor treatment responsivity.

## 1. Introduction

Treatment of recurrent depression in the course of bipolar disorder (BD) is often a challenging clinical task. While classical antidepressants show restricted efficacy and carry a risk of a switch into a manic/hypomanic state [1], not many effective options are available. One such alternative is an anticonvulsant lamotrigine, shown to be efficacious in both treatment and prevention of bipolar depression [2].

The use of lamotrigine poses however its own challenges, not the least being a slow treatment initiation due to the risk of Stevens–Johnson syndrome [3], resulting in a 2–3 month wait before an assessment of clinical response is possible. Given that only a proportion of patients will respond to the drug, this long wait adds to the already high burden of bipolar depression.

Some of this burden could be avoided if the drug were to be allocated only to patients likely to respond to it. A search for biomarkers of treatment response, i.e., biological measures that would indicate who is likely to respond to a given drug, has indeed been one of the key areas in psychiatric research, with some promising findings. For example, in major depressive disorder (MDD), an association of good clinical response to antidepressant treatments with increased activity in the pregenual anterior cingulate cortex (pgACC) has been consistently reported e.g., [4]. Unfortunately, no such data are available for bipolar depression treatments.

The pharmacological mechanism of action of lamotrigine is complex. Among its actions, stabilization of presynaptic neuronal membranes following a reduction in glutamate release is hypothesized to be most relevant to treatment of depression [5]. This may be particularly relevant in the context of depression viewed as a network disorder, in which dysfunctional communication between regions, largely dependent on appropriate glutamate-dependent neuronal communication, is the key to the development and maintenance of symptoms [6].

Over the past two decades, a number of large-scale brain networks have been reliably established, including, among others, the default mode network (DMN), the fronto-parietal network (FPN) and the dorsal attention network (DAN) [7]. This was enabled by the increasingly sophisticated use of functional magnetic resonance imaging (fMRI). A commonly used fMRI based approach is an assessment of resting-state functional connectivity (rsFC), i.e., temporal correlation of activity of the brain structures during the rest, i.e., when an individual is awake yet not engaged in any external tasks. This approach allows detecting spontaneous brain activation patterns that give insight into patterns of neural network activity [8] and allows exploring networks not easily assessed during task activation, such as the DMN. It is free from a task-selection bias and benefits from relatively easy data collection.

Dysfunctional resting state connectivity has been observed in bipolar disorder, suggesting the crucial role of the DMN, frontal regions, as well as the affective and reward networks. A recent meta-analysis, including 1047 bipolar patients and 1081 healthy controls, showed that acute phases of the illness were characterized by altered connectivity of the DMN with the DAN and FPN, and decreased connectivity within the DMN and the affective network [9]. Abnormal connectivity of the DMN, in particular of the medial prefrontal cortex (mPFC), posterior cingulate cortex (PCC) and precuneus, also gained support from a recent systematic review of 23 studies [10]. Another meta-analysis [11] in 494 BD patients and 593 healthy controls (HC), using amplitude of low-frequency fluctuation (ALFF), an approach assessing the intrinsic or spontaneous brain activity, showed decreased ALFF in the bilateral precuneus, left anterior cingulate cortex (ACC), left cerebellum and left superior temporal gyrus (STG), and increased ALFF in frontal regions, bilateral insula, striatum, and right STG.

A few studies explored rsFC changes over the course of treatment, mostly with lithium. One common theme is the normalizing impact of treatments on brain activity. For example, in bipolar depressed patients, a clinical improvement over the course of treatment with lithium was linked to an increase in the general network resilience to disruptions [12], an increase in ALFF in the DMN—in particular in the precuneus and posterior cingulate [13], as well as to normalization of amygdala connectivity [12,14]. In manic patients, both lithium and quetiapine normalized reduced connectivity in the cortico-striatal systems [15]. To our knowledge, no studies on rsFC during lamotrigine treatment have been published.

The present study aimed at identifying differences in patterns of baseline (pre-treatment) resting state functional connectivity in patients with bipolar depression who responded to 10–12 weeks of treatment with lamotrigine, patients who did not respond to such treatment and healthy volunteers.

## 2. Results

### 2.1. Clinical and Demographic Data

Fifteen out of 21 (71%) patients responded to 10–12 weeks of treatment with lamotrigine. Responders and non-responders did not differ with respect to gender, age, baseline depression severity, baseline state anxiety, age at onset and length of illness (see Table 1 for details). Among responders, entering the study, 9 were medicated (i.e., lamotrigine was added to other treatments) and 6 unmedicated (i.e., lamotrigine was the only treatment), and among non-responders—4 were medicated and 2 unmedicated when entering the study. Details of medications taken by individual patients are presented in Appendix A. Healthy controls, as expected, differed from patients in terms of depression and anxiety scores; there were no significant differences in terms of gender and age (Table 1).

### 2.2. Resting-State fMRI Imaging Analysis

Group ICA was restricted to 25 components. Of these, 16 components were identified as RSNs (covering the majority of grey matter (GM) and were evaluated further (Figure 1). These networks corresponded to and overlapped with resting state networks (RSNs) which have been described previously and show high stability over time [7,16,17]. The other components reflected distinct artifacts resulting from scanner/multiband artifacts, head motion and physiological noise (arteries, veins and cerebro-spinal fluid (CSF).

Nonparametric permutation analysis revealed significant group differences in baseline functional connectivity between (1) treatment responders (R) and non-responders (NR); (2) NR and healthy controls (HC); (3) all patients and HC:(1)Treatment R, as compared to NR, showed: greater pre-treatment rsFC between the right FPN and left precuneus; greater pre-treatment FC between the DAN and the left precuneus (Figure 2).(2)Treatment NR, as compared to HC, showed reduced baseline rsFC: of the right FPN with the left precuneus; of the DAN with right middle temporal gyrus (MTG) and left precuneus; of the DMN and left precuneus; of the extended somatosensory-motor area (SSMN) including the sensory-motor network (SMN), auditory cortex, posterior insula, central and parietal operculum, midcingulate cortex (MCC), and supplementary motor area (SMA) with the left hippocampus/left amygdala, left subcallosal cortex/nucleus accumbens, right occipital pole and left middle frontal gyrus (MFG); as well as of the left FPN with left inferior temporal gyrus (ITG)/occipital fusiform gyrus/lateral occipital cortex (Figure 2).(3)All patients, as compared to HC, showed reduced baseline rsFC of the extended sensory-motor area (including the SMN, auditory cortex, posterior insula, central and parietal operculum, MCC, and SMA) with the left hippocampus/left amygdala.

There were no significant differences between R and HC (there were numeric ones).

(1)R > NR:(A)The right fronto-parietal network (FPN) and left precuneus(B)The dorsal attention network (DAN) and left precuneus(2)NR < HC:(A)The right FPN and left precuneus (see 1A);(B)The DAN and left precuneus (see 1B);(C)The DAN and right middle temporal gyrus;(D)The DMN and left precuneus;(E)The left FPN with left inferior temporal gyrus/occipital fusiform gyrus/lateral occipital cortex cluster;(F)The extended sensory-motor component with the left hippocampus/left amygdala (also all patients < HC);(G)The extended sensory-motor component with the left subcallosal cortex/accumbens nucleus cluster.

Details of differences in functional connectivity (temporal correlations) between resting state networks and individual brain regions are presented in Table 2.

Additionally, we presented the correlations between establishes connectivity measures and % change in HAMD scores for all patients (Figure 3), in order to visualize the strength of the correlations. These were statistically significant for all the connections apart from the connection between the left FPN and left inferior temporal gyrus/occipital fusiform gyrus/lateral occipital cortex. Correlations were also done for self-rating depressive questionnaire, Beck Depression Inventory (BDI); as expected, they followed the same pattern. Correlations with anxiety scores assessed with Spielberger Trait Anxiety Inventory (STAI) were not significant. Details of the correlations are presented in Appendix A.

## 3. Discussion

The main aim of this research was to explore baseline (i.e., pre-treatment) patterns of rsFC associated with response to treatment with lamotrigine in patients with bipolar depression. To achieve this, we compared pre-treatment rsFC in bipolar patients with a current episode of depression who responded to 10–12 weeks of treatment with lamotrigine, those who did not respond to such treatment and rsFC in untreated healthy individuals. Using an unbiased ICA approach, we observed an association between good treatment response and increased pre-treatment rsFC, between the right FPN and the left precuneus, and between the DAN and the left precuneus. Additionally, baseline connectivity in future lamotrigine responders did not differ significantly from HC.

Another finding was a more general pattern of lower pre-treatment rsFC in lamotrigine non-responders as compared to responders and healthy controls. In case of the above described connection this difference was significant in comparison to both R and HC. For a number of other connections—between the anterior DMN (mPFC) and the left precuneus; between the left FPN and the left ITG/occipital fusiform gyrus/lateral occipital cortex; and the extended sensory-motor area (including the SMN, auditory cortex, posterior insula, central and parietal operculum, MCC, and SMA) and the left hippocampus/left amygdala, left subcallosal cortex/nucleus accumbens, right occipital pole and left MFG—rsFC was significantly lower in NR as compared to HC, and numerically lower when comparing NR to R. Importantly, no significant differences between R and HC were observed for any of these connections. Additionally, patients as a group differed from HC in reduced baseline rsFC of the sensory-motor component with the left hippocampus, which was however driven by low activity in non-responders and hence will not be discussed in detail.

It can be therefore concluded that patients with bipolar depression, whose resting-state functional connectivity for the above described networks does not differ from healthy controls, are more likely to respond to lamotrigine, with statistically significant differences however present only for the right FPN and DAN, with the left precuneus.

The above findings suggest that preserved rsFC between the part of the DMN—the left precuneus—and both the right FPN and the DAN may be of particular importance to lamotrigine responsivity in bipolar depression. These findings are intriguing given the special relationship between the DMN and the FPN/DAN. The FPN and DAN are often referred to as ‘task positive networks’ (TPNs), which are most active during the execution of external tasks [18]. The FPN, consisting of prefrontal and parietal cortices, is responsible for a flexible and coordinated higher order modulation of cognitive and emotional processes [19,20] and the DAN, consisting of bilateral parietal cortices and frontal eye fields (FEF), is concerned with goal-directed top-down controlled attentional selection and orienting one’s focus to a particular task [21]. The DMN, on the other hand, is a ‘task negative network’ (TNN) [18], primarily involved in inward-directed processing, such as self-referential processing, imagery and memory [22]. The activities of the TPNs and TNN are most often anticorrelated. With some exceptions, DMN’s activity reduces, and TPNs activity increases, when cognitive resources need to be directed towards a task, and this pattern reverses when demand for externally focused resources decreases [18,23].

This is facilitated by rich cortico-cortical connections between the TPNs and DMN, allowing the FPN to exert control over the DMN [20,24,25]. The balance, relying on good communication between the networks, is important for healthy mental function and its changes have been consistently shown in MDD and bipolar disorder [10,26]. Additionally, the importance of both precuneus and the prefrontal cortex (PFC) in the pathophysiology bipolar disorder supported by a recent meta-analysis of whole-brain resting-state functional MRI in 494 bipolar patients and 593 healthy controls [11]. Our study suggests that it may also be relevant for treatment response.

The left precuneus seemed to play a particularly important role in response to lamotrigine in our sample. The DMN consists of distinctive functional subsystems, with the precuneus being one of the hubs of the posterior DMN (pDMN) [22]. Apart from shared DMN functions, the pDMN is specifically involved in involuntary (bottom-up) attentional processing and shows a bias towards self-centered rather than self-other processing. The precuneus has particularly rich connections with both the FPN and DAN and its connectivity may have a particular role in maintaining the balance between the TPN and TNN [20]. Research in major depressive disorder suggested a key role of the pDMN in antidepressant response [27,28,29].

While the main question of this research was whether responders and non-responders to lamotrigine differed in terms of baseline rsFC, we also we observed a more general pattern of lower rsFC in lamotrigine NR across a number of networks. In addition to above discussed connections, for which rsFC was significantly lower in NR as compared to both R and HC, we saw a number of connections for which connectivity in NR was reduced significantly when compared to HC, and numerically when compared to R, and for which R did not differ from HC. This was seen for rsFC between the mPFC and the left precuneus—the hubs of, respectively, the anterior and posterior DMN [22], suggesting that ineffective communication within the DMN may be associated with the NR status.

This finding points at a more widespread reduction in the left precuneus functional connectivity in lamotrigine non-responders, with rsFC with the DAN and FPN being relevant to treatment response. Poor lamotrigine responsivity was also associated with decreased connectivity of the extended sensory-motor area including, mostly primary, somatosensory and somatomotor areas (the SMN, auditory cortex, posterior insula, central and parietal operculum, MCC, and SMA), with the regions involved in an array of emotional, cognitive and reward functions (the left hippocampus, left amygdala, left subcallosal cortex, left nucleus accumbens, left MFG and right occipital pole). Despite its structural complexity, the sensory-motor component can be defined through its regions’ common function, sensory and motor processing. The importance of accurate perception and integration of internal and external sensory data for the development of emotional states [30] and the subjective experience of emotion [31,32,33] is widely accepted. Decreased connectivity of the sensory-motor regions is therefore likely to impact emotional/cognitive processing, which may in turn hinder medication response. Similarly, in NR we saw decreased rsFC of the left FPN with the left fusiform gyrus and lateral occipital cortex, the regions essential for the processing of salient emotional information deriving from faces and recognition of shapes [34].

The correlations between established connectivity measures and percent change in HAMD scores for all patients were additionally presented in Figure 3. The main reason was to provide a visualization of the effect in individual patients in the measures already established through the whole-brain analysis. However, while interpreting these, in order to avoid the pitfalls of circularity, it needs to be remembered that the data presented were extracted from the regions that survived a multiple comparisons analysis.

To our knowledge, there are no published studies exploring the relevance of resting state connectivity to response to lamotrigine. In fact, not much research has been conducted into the relationship between resting-state brain activity and response to pharmacological treatments in bipolar disorder. The majority of studies used lithium, mainly in manic/hypomanic patients, and focused on changes in resting-state brain activity over the course of treatment, e.g., [12,14,15]. While they did not explicitly explore the baseline markers of future treatment response, they provided an insight into the mechanisms of drug action, and some showed correlations between changes in brain activity and clinical improvement. Despite their undoubted value, the nature of these findings makes comparisons with our study somehow limited.

The effect of lamotrigine on brain activity has been somehow more widely explored in the context of emotional and cognitive tasks. Some studies suggested a link between symptomatic improvement and normalization of activity in a number of brain regions, including the PFC [35,36,37,38,39]. Most of these studies included children and adolescents, which is different to our study. In any case, comparisons between task-evoked and resting state activity need to be approached with caution, as the two paradigms probe different networks and their relationships, e.g., task fMRI has a limited value when assessing the DMN activity. It is however interesting that in the task studies clinical improvement was correlated with normalization of the brain activity and that in our study good treatment response was shown in patients with preserved rsFC.

This study has both strengths and limitations. An important strength was an inclusion of patients with bipolar depression only, while previous studies often assessed mixed groups of patients in hypomanic, manic, mixed or depressed states. The group was well-characterized, and the patients underwent assessments by the same psychiatrist to increase homogeneity. Treatment with lamotrigine followed clinical guidelines, and although it was an open-label approach driven by clinical needs, in all patients but two, their final dose was 200 mg daily. An important strength was the analytical methods chosen, ICA, which allows for an unbiased exploration of functional networks. It is not restricted by hypotheses, contrary to the seed-based approaches. Stringent statistical methods, nonparametric permutation testing and threshold-free cluster enhancement (TFCE), were used to increase confidence in the results. At the same time, it is important to remember that this is an exploratory study and therefore correction for multiple comparisons (inclusion of multiple resting state networks) was not performed. The main limitation is the small size of the group, which increases the likelihood of type 2 error and decreases the power. This means that some potentially important findings might have fallen under the threshold of detection. Another potential caveat is that some patients were treated with mood-stabilizers/antipsychotics and/or antidepressants when entering the study, while some were untreated at this point. This study was designed to reflect the clinical practice and, importantly, all patients were depressed at the time of the study; hence, it can be assumed that the impact of the current treatments was limited. However, the fact that both medicated and unmedicated patients were included needs to be taken into account when interpreting the results. The impact of medications on MRI indices has been poorly studied as far, in particular in terms of its impact on resting state activity. This however reflects the clinical practice in which lamotrigine is often an add-on medication to other drugs. A further limitation of this work is the lack of longitudinal assessment of rsFC, i.e., exploration of mechanisms of lamotrigine action. The focus of this paper was however to explore potential indices of clinical response to lamotrigine in the context of clinical feasibility, hence it presents the results based on a single MRI scan. Future studies exploring the subject of mechanisms of action in the context of rsFC, at the same time providing a replication sample, will be needed. One general limitation, shared by all functional connectivity studies, is related to the very nature of this approach, where the estimation of the functional connectivity is based on the temporal correlation of activities in various brain regions, and does not provide information about anatomical connections nor directionality of the effect or causality. Hypotheses regarding the meaning of findings may however be proposed based on known anatomy and function, as well as interactions of structures involved. Also, it needs to be noted that this study did not assess prediction of response to lamotrigine, for which a different type of the analysis, e.g., a leave-one-out (LOO) approach, would be necessary. This type of analysis was deemed to be unreliable due to a small sample size. Instead, we explored associations between patterns of rsFC and future response to lamotrigine.

To summarize, the results of our study suggest that preserved rsFC between the FPN and DAN, the networks involved in cognitive control, and the hub of the posterior DMN, the left precuneus, may be critical for good response to lamotrigine as an add-on treatment in patients with bipolar depression. The study also suggested a more general decrease in rsFC to be related to poor treatment responsivity. Further work on predictors of response to lamotrigine is warranted. An identification of such predictors could largely support the clinical practice and save patients from long wait periods until the drug efficacy can be assessed. This pilot study creates promising grounds for further research into this important clinical subject.

## 4. Materials and Methods

### 4.1. Participants and Design

Twenty-nine patients with bipolar disorder according to DSM-IV were recruited into the study, 16 females (mean age 33 years, range 19–56) and 13 males (mean age 36 years, range 22–58). All participants were assessed with the Structured Clinical Interview for DSM-IV [40] for the presence of current and past psychiatric disorders. Fifteen patients had the diagnosis of bipolar type 1, and fourteen patients bipolar type 2. All patients also met criteria for Major Depressive Episode. Twenty healthy subjects were recruited into the control group, 13 females (mean age 30 years, range 19–58) and 7 males (mean age 32 years, range 19–51). Exclusion criteria specific for patients were: psychosis or substance dependence as defined by DSM-IV and contraindications to lamotrigine treatment or lamotrigine not considered the best next line of treatment; for healthy controls: any psychiatric disorders as defined by DSM-IV; for both groups: contra-indications to MR imaging and concurrent medication which could alter emotional processing (e.g., benzodiazepines). The study was approved by the Research Ethics Committee South Central—Hampshire A and performed in accordance with Declaration of Helsinki.

Patients were started on lamotrigine by their treating clinician. Treatment was applied according to clinical guidelines and patient’s needs. Of the 24 patients who were scanned at baseline, in nine lamotrigine was their only drug therapy; in the other fifteen, lamotrigine was added to current drug treatments, including sertraline, citalopram, fluoxetine, venlafaxine, duloxetine, mirtazapine, quetiapine, aripiprazole, lithium, and sodium valproate. This drug treatment had been unchanged for at least 6 months before patients entered the study and remained unchanged during the study. The details of treatments for individual patients are presented in Appendix A.

All patients and controls underwent an MRI scan—in case of patients, before lamotrigine treatment started. Clinical response was assessed after 10–12 weeks.

Clinical response to lamotrigine treatment was assessed by the Hamilton Rating Scale for Depression (HAM-D). HAM-D scores were measured at baseline and at the follow-up session after 10–12 weeks of treatment. Response to treatment was defined as a 50% decrease in HAMD score from baseline [41]. Manic/hypomanic scores were measured by Altman Self-Rating Mania Scale [42], and anxiety levels were measured with Spielberger’s State-Trait Anxiety Inventory [43]—both at baseline and after 10–12 weeks of treatment. Participants also rated their depression by Beck Depression Inventory (BDI) [44]. Healthy control subjects followed the same protocol, however, they did not receive any medication, and had only baseline ratings.

Baseline resting state fMRI data and response status were available for twenty-one patients. Five patients were unable to complete the baseline fMRI protocol due to high levels of anxiety in the scanner (four patients) and restless legs syndrome (one patient). Three patients did not return for their follow-up assessment: one where lamotrigine was stopped by their treating clinician, one due to an eye surgery, and one because of practical issues.

### 4.2. Resting-State fMRI

Scanning was performed at the University of Oxford Centre for Clinical Magnetic Resonance Research (OCMR), using a 3-Tesla Siemens scanner with a 32-channel head-coil. The neuroimaging protocol comprised functional and structural sequences as follows.

Whole-brain resting-state imaging was performed using a gradient-echo EPI sequence (TR = 2000 ms, TE = 28 ms, flip angle = 89°, field of view = 192 mm, voxel dimension = 3 × 3 × 3.5 mm^3^ isotropic, 180 volumes, acquisition time = 6 min 4 s). Participants were instructed to stay still, to keep their eyes closed, think of nothing in particular, and not to fall asleep. Images were distortion corrected by an acquired fieldmap (echos at 5.19 and 7.65 ms, TR = 488ms, flip angle = 60°).

Structural scans were acquired via T1-weighted MR images (TR = 2040 ms, TE = 4.7 ms, flip angle = 8°, field of view = 192 mm, voxel dimension = 1mm isotropic, acquisition time = 5 min 56 s). Participants were again allowed to close their eyes but instructed not to fall asleep.

### 4.3. Analysis Methods

Data were analyzed using FSL (FMRIB Software Library v6.6) tools (https://fsl.fmrib.ox.ac.uk/fsl, accessed on 9 April 2021). ICA analysis was unrestrained at subject level and restrained to 25 components at group level.

#### 4.3.1. Pre-Processing

The structural anatomical scans were brain extracted using FSL’s Brain Extraction Tool (BET) [45]. Single subject functional MRI data were pre-processed and analyzed using Multivariate Exploratory Linear Decomposition into Independent Components (Melodic, part of FSL). Pre-processing involved a number of steps designed to reduce unwanted variability in the data and to improve the validity of statistical analysis. The following steps where therefore implemented for each subject: (1) Removal of non-brain structures using BET [45] motion correction using FMRIB’s Linear Image Registration Tool (MCFLIRT; [46], (2) distortion correction (fsl_prepare_fieldmap was used to generate the required input data), (3) spatial smoothing using a Gaussian kernel of FWHM 6mm, (4) grand-mean intensity normalization of the entire 4D dataset by a single multiplicative factor and (5) high-pass temporal filtering cut-off = 150s (Gaussian-weighted least-squares straight line fitting, with sigma = 75.0 s). Furthermore, resting state specific pre-processing steps included masking of non-brain voxels; voxel-wise de-meaning of the data and normalization of the voxel-wise variance. In addition, functional images were registered to their high resolution structural via the high contrast functional image and Boundary-Based Registration (BBR) using FLIRT [46,47], and non-linear registration from structural to MNI [Montreal Neurological Institute (MNI)] standard space was then further refined using FNIRT nonlinear registration [48,49], resampling resolution = 2 mm.

#### 4.3.2. ICA Analysis

Probabilistic ICA [50] implemented using FSL’s MELODIC was used to analyze the resting state data.

First, pre-processed data were whitened and projected into a dimensional subspace (actual dimension subject dependent) using probabilistic Principal Component Analysis (PCA) where the number of dimensions was estimated using the Laplace approximation to the Bayesian evidence of the model order [50,51].

The whitened observations were decomposed into sets of vectors which describe signal variation across the temporal domain (time-courses) and across the spatial domain (maps) by optimizing for non-Gaussian spatial source distributions using a fixed-point iteration technique [52]. Estimated Component maps were divided by the standard deviation of the residual noise and thresholded by fitting a mixture model to the histogram of intensity values [50].

Resting-state data were then denoised by ICA denoising using FMRIB’s ICA-based Xnoiseifier (FIX) by manually creating a training dataset from five subjects in each group (N = 10) randomly chosen by BG. Manual labeling of components was done by MM and BG and then compared. A conservative approach was used in that if not agreed the component was included.

The pre-processed cleaned functional data were then temporally concatenated across subjects in order to create a single 4D dataset. Then, the (group-wise) concatenated multiple rs-fMRI datasets were decomposed using a group ICA to identify large-scale patterns of FC in the population of subjects (restrained to 25 components) as described previously [53]. Components corresponding to known RSNs were identified by eye and compared to previously published maps [7] using Pearson spatial cross-correlation.

The set of spatial maps from this group-average analysis was used to generate subject-specific versions of the spatial maps, and associated time series, using dual regression [16,53]. This is a two-step process. First, for each subject, the group-average set of spatial maps is regressed (as spatial regressors in a multiple regression) into the subject’s 4D space-time dataset. This results in a set of subject-specific time series, one per group-level spatial map. Next, those time series are regressed (as temporal regressors, again in a multiple regression) into the same 4D dataset files (1 per original ICA map, with the fourth dimension being subject identification), resulting in a set of subject-specific spatial maps, one per group-level spatial map. These were then tested by a voxel-wise GLM based analysis for statistically significant differences between groups using nonparametric permutation testing (5000 permutations) using FSL’s randomize permutation-testing tool (Randomise). Cluster-based thresholding was applied, using TFCE and a family-wise-error corrected cluster significance threshold of *p* < 0.05 applied to the suprathreshold clusters. For completeness (and because previous studies also used less stringent statistical thresholds), uncorrected images with *p* < 0.005 are also shown. GM maps were added as voxel-wise covariates (see below).

The GLM comparison included the groups of interest comparison (responders vs. non-responders, mean responders non-responders vs. HC). To further visualize the results, individual Parameter Estimates (PE) values were extracted from their custom maps, using significant clusters as binary masks.

GM images of each subject were extracted using Feat_GM_prepare and added to the model used to analyze the functional data to remove any potential structural differences explaining the BOLD contrast differences.

All activations are reported using MNI coordinates.

## Figures and Tables

**Figure 1 pharmaceuticals-14-00534-f001:**
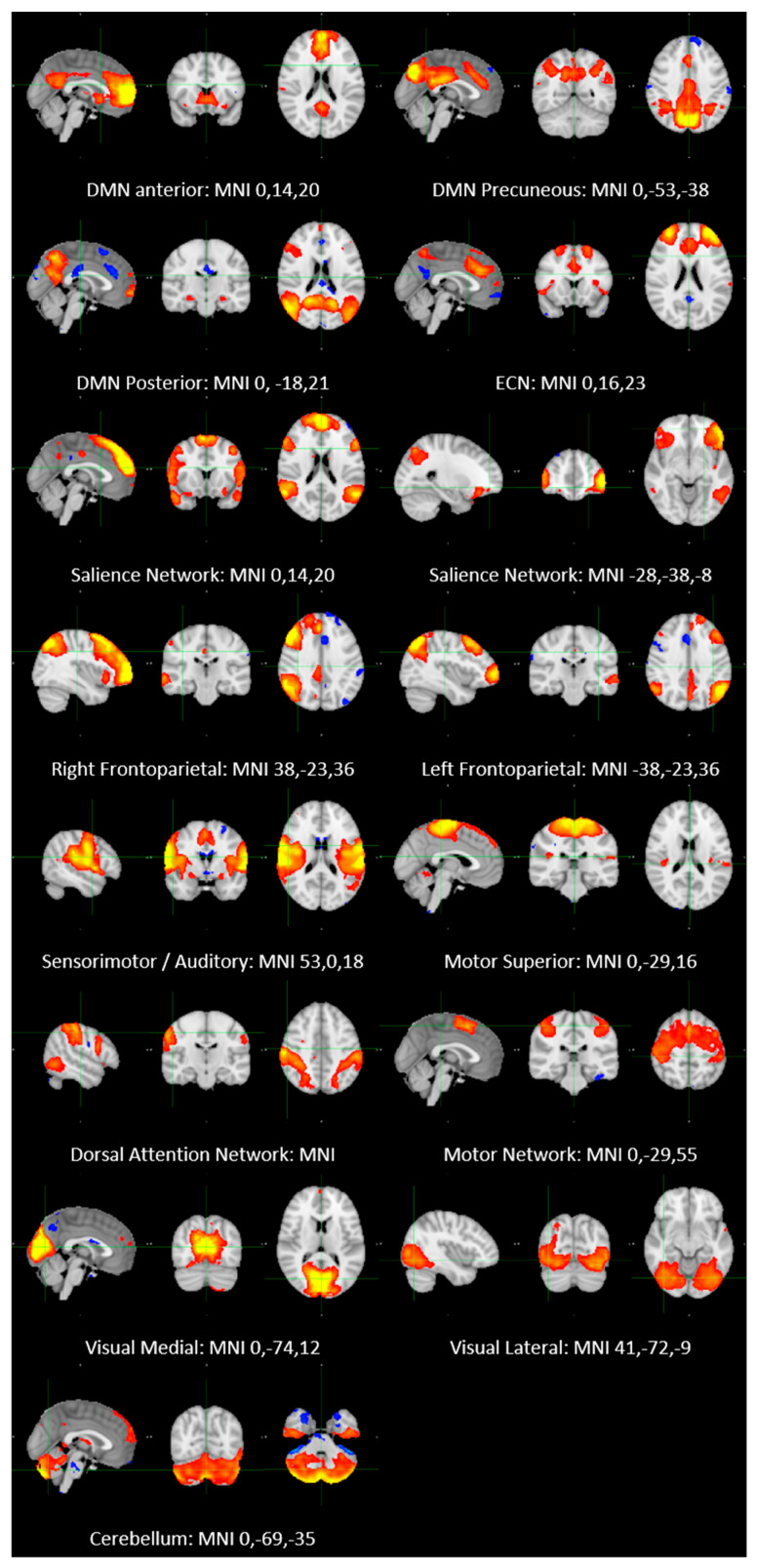
Resting state networks identified in the study. Axial, coronal and sagittal slices for the main resting state networks detected, overlaid onto the standard MNI brain. All maps were thresholded at *Z* = 3.

**Figure 2 pharmaceuticals-14-00534-f002:**
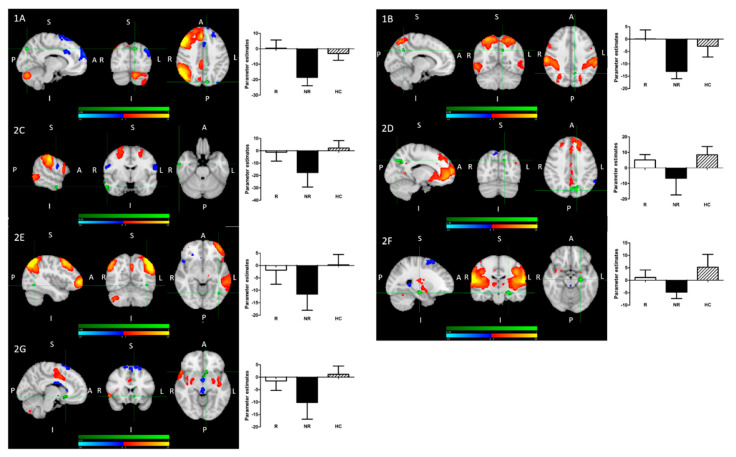
Pre-treatment resting-state functional connectivity in treatment responders, non-responders and healthy controls. Significantly greater temporal correlation (coherence) of activity between individual networks (red-yellow) and clusters (green), differentiating the groups, is shown in the left panels. The networks and regions are described in detail below and in Table 2. Parameter estimates, representing the value of temporal correlation for individual groups, are shown on the graphs directly to the right of the corresponding brain map.

**Figure 3 pharmaceuticals-14-00534-f003:**
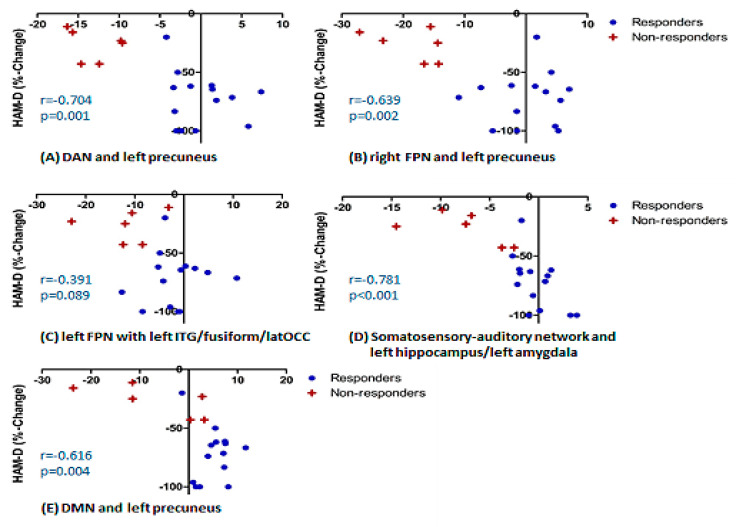
Correlations between established connectivity measures and % change in HAMD scores. DAN-dorsal attention network; FPN-fronto-parietal network; DMN-default mode network; ITG-inferior temporal gyrus; latOCC-lateral occipital cortex.

**Table 1 pharmaceuticals-14-00534-t001:** Demographics and clinical scores (for patients for whom both valid baseline and follow-up resting state data were available), presented as a mean ± standard deviation (SD). Treatment responders (R) and non-responders (NR) differed significantly at the second scan assessment in HAMD score (*t* = −4.140, *p* = 0.001) and BDI score (*t* = −3.994, *p* = 0.001). All other differences between R and NR, as well as R, NR and healthy controls (HC), were not significant (*p* > 0.05).

	Responders *N* = 15	Non-Responders *N* = 6	Healthy *N* = 20
Age (years)	34.8 ± 11.1	29.8 ± 11.3	30.7 ± 10.0
Gender	11 F and 4 M	3 F and 3 M	13 F and 7 M
Age at onset (years)	20.1 ± 10.1	15.5 ± 5.3	NA
Length of illness (years)	14.1 ± 6.3	14.3 ± 10.0	NA
HAMD baseline	20.8 ± 8.3	18.5 ± 4.5	0.5 ± 0.9
HAMD at 2nd scan	5.5 ± 3.9	13.8 ± 4.8	NA
BDI baseline	24.3 ± 12.7	26.0 ± 7.1	0.9 ± 1.6
BDI at 6 weeks	6.5 ± 5.2	18.8 ± 8.9	NA
Altman baseline	2.7 ± 2.8	3.2 ± 6.3	0 ± 0
Altman at 2nd scan	4.27 ± 4.4	5.3 ± 3.5	NA
State anxiety	34.1 ± 10.7	42.8 ± 16.5	27.9 ± 8.4

**Table 2 pharmaceuticals-14-00534-t002:** Differences in functional connectivity (temporal correlations) between resting state networks and individual brain regions. Individual clusters identified with each contrast shown. Resting state data TFCE-corrected FWE cluster significance level of *p* < 0.05.

Contrast	Network	Cluster	Cluster Size (Number of Voxels)	Peak Voxel (MNI)	1-Pmax Value
R > NR	Right fronto-parietal network	Left precuneus cortex	43	−10,−80,40	>0.999
	Dorsal attention network	Left precuneus cortex	27	−10,−58,36	0.983
NR < HC	Somatosensory-motor network	Left hippocampus	112	−24,−18,−16	0.990
		Left subcallosal cortex, accumbens	93	−8,18,−8	0.979
		Right occipital pole	36	8,−102,4	0.971
	Default Mode Network	Left precuneus cortex	208	14,−74,38	0.992
	Right fronto-parietal network	Left precuneus cortex	39	−10,−80,40	0.997
	Left fronto-parietal network	Left inferior temporal gyrus/occipital fusiform gyrus/lateral occipital cortex	17	−40,−62,−8	0.967
	Dorsal attention network	Right middle temporal gyrus	51	60,−6,26	0.983
		Left precuneus cortex	31	−10,−58,34	0.986
		Right middle temporal gyrus	11	64,−6,−16	0.962
HC > R + NR	Somatosensory-motor network	Left hippocampus	15	−24,−18,−16	0.966

## Data Availability

The data presented in this study are available on request from the corresponding author.

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
