# Peer review of "Functional Connectivity between Task-Positive Networks and the Left Precuneus as a Biomarker of Response to Lamotrigine in Bipolar Depression: A Pilot Study"

_pharmaceuticals, 2021, doi:10.3390/ph14060534_

Round 1
Reviewer 1 Report
This is an important manuscript involving functional connectivity to evaluate the neural response to lamotrigine in bipolar depression. Here is a suggestion to help improve the manuscript:
1) The methods and figures showing the functional connectivity networks using resting-state are just excellent. The authors thoroughly pre-processed and identified several very important networks in the brain. My main problem is that I don’t like artificially assigning a threshold to define the responders and non-responders of the therapy. It would be better to do a correlational analysis where the Hamilton Depression Rating Scale (HAMD) score could be correlated with a measure of the functional connectivity comparing the treatment group with the control group instead of comparing responders versus non-responders. The authors have already developed excellent functional connectivity objective measures which could be used in this correlational analysis. To be clear, the correlation would be between the HAMD score and the functional connectivity measure within the treatment group.
Author Response
We would like to thank the Reviewer for their assessment of our manuscript. We hope that the changes will be satisfactory.
This is an important manuscript involving functional connectivity to evaluate the neural response to lamotrigine in bipolar depression. Here is a suggestion to help improve the manuscript:
The methods and figures showing the functional connectivity networks using resting-state are just excellent. The authors thoroughly pre-processed and identified several very important networks in the brain. My main problem is that I don’t like artificially assigning a threshold to define the responders and non-responders of the therapy. It would be better to do a correlational analysis where the Hamilton Depression Rating Scale (HAMD) score could be correlated with a measure of the functional connectivity comparing the treatment group with the control group instead of comparing responders versus non-responders. The authors have already developed excellent functional connectivity objective measures which could be used in this correlational analysis. To be clear, the correlation would be between the HAMD score and the functional connectivity measure within the treatment group
We agree that using the 50% threshold for response is somehow arbitrary. We decided to implement it so that our study can be compared with other studies exploring the subject of response, and commonly using this threshold. We however fully agree with the Reviewer that correlations may be more informative and we included the figure showing the correlation between improvement in depression, measured as % change in the initial HAMD scores and functional connectivity measures. We also provided a supplementary table including the details of the correlations with Beck Depression Inventory (BDI) scores, and Spielberger Trait Anxiety Inventory (STAI).
We added the following fragment to the Results section:
‘Additionally, we presented the correlations between establishes connectivity measures and % change in HAMD scores for all patients (Figure 3), in order to visualize the strength of the correlations. These were statistically significant for all the connections apart from the connection between the left FPN and left inferior temporal gyrus/occipital fusiform gyrus/lateral occipital cortex. Correlations were also done for self-rating depressive questionnaire, Beck Depression Inventory (BDI); as expected, they followed the same pattern. Correlations with anxiety scores assessed with Spielberger Trait Anxiety Inventory (STAI) were not significant. Details of correlations are presented in Supplementary Table 2.’
In order to avoid the criticism of circular analysis, we added the following fragment to the Discussion:
‘The correlations between established connectivity measures and percent change in HAMD scores for all patients were additionally presented in Figure 3. The main reason was to provide a visualization of the effect in individual patients in the measures al-ready established through the whole-brain analysis. However, while interpreting these, in order to avoid the pitfalls of circularity, it needs to be remembered that the data presented were extracted from the regions that survived a multiple comparisons analysis.’
Reviewer 2 Report
The authors presented a study focused on neuronal underpinnings of response to lamotrigine in patients diagnosed with bipolar disorder. In general, the aim of the study concerning relationships between functional connectivity and the possibility of individualizing pharmacotherapy is fully justified both from a purely scientific and clinical point of view. Despite this undoubtfully well-justified goal, the study and the manuscript itself suffer from several shortcomings needing substantial revision.
- there is a problem throughout the whole manuscript with consistency regarding neuroanatomical nomenclature. The problem is the use of abbreviations for the names of anatomical structures, which are largely unexplained, besides, the authors are constantly changing the rules by giving some names in full and others only abbreviations that are often not explained in the text. As a result, in many places the text is incomprehensible, and in some parts, it is self-contradictory. An example is a sentence on page 2 (lines 80-83): "Another meta-analysis [11] in 494 BD patients and 593 HC, using amplitude of low-frequency fluctuation (ALFF), an approach assessing the intrinsic or spontaneous brain activity, showed decreased ALFF in the precuneus, ACC, cerebellum and STG, and increased ALFF in frontal regions, insula, striatum, and superior temporal gyrus (STG)." In the first half of the sentence, something called STG showed decreased ALFF, and in the other half of the sentence superior temporal gyrus (STG) showed increased ALFF. It is absolutely necessary to standardize the rules for giving abbreviations, and above all to give the names of neuroanatomical structures in their full wording.
- the inclusion of patients receiving only lamotrigine and patients receiving lamotrigine together with a very large number of other substances with completely different pharmacological effects, in fact, limits any conclusions regarding neuronal underpinnings of lamotrigine treatment and response. Please, explain in full details how did you separate the therapeutic effect of lamotrigine itself from the effect of lamotrigine in polytherapy with other drugs? It is necessary to verify obtain results separately in subgroups of patients receiving only lamotrigine from other patients treated with polytherapy.
- how many responders and non-responders received other drugs than lamotrigine?
- in papers focused on brain connectivity figures should present data showing connectivity, not only separated structures that were co-activated during scanning. Figure 2 should present LINKS between synchronized structures, with some graphical specificity of hypo- and hyper-connections in compared subgroups. In the current form, figures look like showing typical results of fMRI scanning without connectivity-related data.
- In general, this study seems to have been carried out only until its halfway. If the study assessing the psychopathological dimension of patients' functioning in terms of their improvement and response to lamotrigine treatment was conducted twice, i.e. before the drug introduction and after a period of 10-12 weeks, the reader has the right to expect that the fMRI assessment will also be performed twice, analogically, before treatment and after 10-12 weeks, thus showing how the patterns of neural synchronization have changed as a result of the treatment. I am aware that now it is probably no longer possible to reassess patients with fMRI, but the lack of a second evaluation leaves many questions unanswered. This is a limitation of the presented research.
Author Response
We would like to thank the Reviewer for their assessment of our manuscript. We hope that the changes will be satisfactory.
The authors presented a study focused on neuronal underpinnings of response to lamotrigine in patients diagnosed with bipolar disorder. In general, the aim of the study concerning relationships between functional connectivity and the possibility of individualizing pharmacotherapy is fully justified both from a purely scientific and clinical point of view. Despite this undoubtfully well-justified goal, the study and the manuscript itself suffer from several shortcomings needing substantial revision.
- there is a problem throughout the whole manuscript with consistency regarding neuroanatomical nomenclature. The problem is the use of abbreviations for the names of anatomical structures, which are largely unexplained, besides, the authors are constantly changing the rules by giving some names in full and others only abbreviations that are often not explained in the text. As a result, in many places the text is incomprehensible, and in some parts, it is self-contradictory. An example is a sentence on page 2 (lines 80-83): "Another meta-analysis [11] in 494 BD patients and 593 HC, using amplitude of low-frequency fluctuation (ALFF), an approach assessing the intrinsic or spontaneous brain activity, showed decreased ALFF in the precuneus, ACC, cerebellum and STG, and increased ALFF in frontal regions, insula, striatum, and superior temporal gyrus (STG)." In the first half of the sentence, something called STG showed decreased ALFF, and in the other half of the sentence superior temporal gyrus (STG) showed increased ALFF. It is absolutely necessary to standardize the rules for giving abbreviations, and above all to give the names of neuroanatomical structures in their full wording.
We do apologise for the lack of consistency regarding neuroanatomical nomenclature, rightly brought up by the Reviewer as an issue to correct. We have edited the manuscript explaining the abbreviations when the name first showed in the manuscript and subsequently only used abbreviations. We apologise in particular for the fragment cited above - the confusion stemmed from our omission of ‘left and ‘right’ when talking about the specific structures, which has now been corrected. We hope this provides the manuscript with necessary consistency.
- the inclusion of patients receiving only lamotrigine and patients receiving lamotrigine together with a very large number of other substances with completely different pharmacological effects, in fact, limits any conclusions regarding neuronal underpinnings of lamotrigine treatment and response. Please, explain in full details how did you separate the therapeutic effect of lamotrigine itself from the effect of lamotrigine in polytherapy with other drugs? It is necessary to verify obtain results separately in subgroups of patients receiving only lamotrigine from other patients treated with polytherapy.
Thank you for raising these very important points. We allowed for an inclusion of both unmedicated patients and those treated with different medications as the study followed a naturalistic study reflecting the ‘real-life’ clinical practice. Indeed, patients were treated with mood stabilizers/antipsychotics, but also antidepressants, and some were unmedicated. The goal of this pilot study was to identify preliminary measures of response to lamotrigine, which would be useful for clinicians, and which subsequently could be explored in more detail in future, larger studies. In medicated patients, depression developed despite them remaining on a stable dose of other medications for at least 6 months, hence it can be assumed that the impact of the treatments was limited.
In the Results section, we added:
‘Among responders, entering the study, 9 were medicated (i.e. lamotrigine was added to other treatments) and 6 unmedicated (i.e. lamotrigine was the only treatment), and among non-responders - 4 were medicated and 2 unmedicated when entering the study. Details of medications taken by individual patients are presented in the Supplementary Table 1.’
As this issue is an important issue, we addressed it in more detail in the manuscript. We provided a supplementary table detailing medication use in each subject.
In the Methods section we added:
‘This drug treatment had been unchanged for at least 6 months before patients entered the study and remained unchanged during the study. The details of treatments for individual patients are presented in Supplementary Table 1.’
Unfortunately running the analysis including only unmedicated or medicated patients would result in insufficient power due patient distribution (unmedicated: 6 responders and 2 non-responders; medicated: 9 responders and 4 non-responders) and while acknowledging validity of the point raised by the Reviewer, we are unable to provide separate analysis for unmedicated and medicated patients.
We addressed the above issues in the Limitations section:
‘Another potential caveat is that all some patients were treated with different mood-stabilizers/antipsychotics and/or antidepressants when entering the study, while some were untreated at this point. This study was designed to reflect the clinical practice and, importantly, all patients were depressed at the time of the study, hence it can be assumed that the impact of the current treatments was limited. However, the fact that both medicated and unmedicated patients were included needs to be taken into account when interpreting the results.’
- how many responders and non-responders received other drugs than lamotrigine?
We provided a supplementary table detailing medications taken by each individual patient, together with their response status.
- in papers focused on brain connectivity figures should present data showing connectivity, not only separated structures that were co-activated during scanning. Figure 2 should present LINKS between synchronized structures, with some graphical specificity of hypo- and hyper-connections in compared subgroups. In the current form, figures look like showing typical results of fMRI scanning without connectivity-related data.
Thank you very much for this comment, which made us realize that the results were not presented as clearly as we wished. We have corrected it by changing the figure caption and making sure that the colour code indicating functional connections between the networks and clusters significantly different between the groups can be easily interpreted (i.e. red-yellow – the network of interest in positive correlation, blue – the network of interest in negative correlation, and green – cluster where differences between the groups in temporal correlation with the network in question has been observed). The Reviewer is right that the picture of the brain does not show the direction of between-group differences, only that the groups differed in terms of these connections. The method of the resting state analysis we used, independent component analysis (ICA), is a multi-varied voxel-based spatial approach. It only allows to observe which parts of the brain co-activate, which is typically presented on the picture of the brain using a colour code, without the links between the structures, which would be more typical for node-based temporal approaches. For this reason, we decided to leave the pictures unchanged. Important information about group differences is shown in detail in the figures placed directly by the picture of the brain.
We change the caption to Figure 2 as follows:
‘Significantly greater temporal correlation (coherence) of activity of individual networks (red-yellow) and clusters (green), differentiating the groups, is shown in the left panels. The networks and regions are described in detail below and in Table 2. Parameter estimates, representing the value of temporal correlation for individual groups, are shown on the graphs directly to the right of the corresponding brain map.’
Using the example of rsFC between the right fronto-parietal network, the way to read the figures, using both the picture of the brain and the graph, as well as information provided in Table 2, is as follows:
In responders, compared to non-responders, significantly greater resting state functional connectivity (i.e. temporal correlation), was observed between the right fronto-parietal network (FPN) and a cluster in the left precuneous (in green, p<0.001, peak voxel location: x=-10, y=-80, z=40, cluster size=43 voxels).
- In general, this study seems to have been carried out only until its halfway. If the study assessing the psychopathological dimension of patients' functioning in terms of their improvement and response to lamotrigine treatment was conducted twice, i.e. before the drug introduction and after a period of 10-12 weeks, the reader has the right to expect that the fMRI assessment will also be performed twice, analogically, before treatment and after 10-12 weeks, thus showing how the patterns of neural synchronization have changed as a result of the treatment. I am aware that now it is probably no longer possible to reassess patients with fMRI, but the lack of a second evaluation leaves many questions unanswered. This is a limitation of the presented research.
Thank you for this comment. We agree this is a limitation of the current paper and added a line to the limitation section of the discussion stating this. It would be indeed very interesting to explore mechanisms of lamotrigine action in more detail. At the same time, this particular paper may benefit from keeping the message simple and focused on what can be useful in the clinical context, i.e. who is likely to respond to lamotrigine based on a single MRI scan. Of course, this will require replication and prediction analysis in a bigger sample.
We added the following fragment to the Discussion:
‘A further limitation of this work is the lack of longitudinal assessment of rsFC, i.e. exploration of mechanisms of lamotrigine action. The focus of this paper was however to explore potential indices of clinical response to lamotrigine in the context of clinical feasibility, hence it presents the results based on a single MRI scan. Future studies exploring the subject of mechanisms of action in the context of rsFC, at the same time providing a replication sample, are needed.’
Round 2
Reviewer 1 Report
I feel that the authors have addressed my previous concerns and the article is now ready for publication.
Reviewer 2 Report
The authors fully responded to the recommendations and suggestions and significantly improved the manuscript.